# Modeling Adsorption, Conformation, and Orientation of the Fis1 Tail Anchor at the Mitochondrial Outer Membrane

**DOI:** 10.3390/membranes12080752

**Published:** 2022-07-31

**Authors:** Beytullah Ozgur, Cory D. Dunn, Mehmet Sayar

**Affiliations:** 1College of Engineering, Koç University, Sarıyer, İstanbul 34450, Turkey; bozgur@ku.edu.tr; 2Institute of Biotechnology, University of Helsinki, 00014 Helsinki, Finland; 3Chemical & Biological Engineering Department, College of Engineering, Koç University, Sarıyer, İstanbul 34450, Turkey; 4Mechanical Engineering Department, College of Engineering, Koç University, Sarıyer, İstanbul 34450, Turkey

**Keywords:** peptide conformation, mitochondria, mitochondrial outer membrane, tail anchor, molecular dynamics, lipid membrane

## Abstract

Proteins can be targeted to organellar membranes by using a tail anchor (TA), a stretch of hydrophobic amino acids found at the polypeptide carboxyl-terminus. The Fis1 protein (Fis1p), which promotes mitochondrial and peroxisomal division in the yeast *Saccharomyces cerevisiae*, is targeted to those organelles by its TA. Substantial evidence suggests that Fis1p insertion into the mitochondrial outer membrane can occur without the need for a translocation machinery. However, recent findings raise the possibility that Fis1p insertion into mitochondria might be promoted by a proteinaceous complex. Here, we have performed atomistic and coarse-grained molecular dynamics simulations to analyze the adsorption, conformation, and orientation of the Fis1(TA). Our results support stable insertion at the mitochondrial outer membrane in a monotopic, rather than a bitopic (transmembrane), configuration. Once inserted in the monotopic orientation, unassisted transition to the bitopic orientation is expected to be blocked by the highly charged nature of the TA carboxyl-terminus and by the Fis1p cytosolic domain. Our results are consistent with a model in which Fis1p does not require a translocation machinery for insertion at mitochondria.

## 1. Introduction

Biochemical, genetic, and computational approaches have significantly contributed to our understanding of the signals that direct proteins to membranes. A number of polypeptides are directed to organelles with the assistance of a carboxyl-terminal, hydrophobic “tail anchor” (TA). Nearly 5% of proteins are targeted to a destination membrane by a TA [1], and new tail-anchored proteins continue to be identified [2]. However, the mechanisms by which TAs are targeted to and inserted at specific subcellular locations remain poorly defined [3,4,5,6,7,8].

The Fis1 protein (Fis1p) recruits additional factors to the surface of mitochondria and peroxisomes to promote their division in *Saccharomyces cerevisiae* [9,10,11,12]. Fis1p consists of a cytosolic domain that binds to its partner proteins, as well as a TA required for tethering this polypeptide to organelle surfaces [13,14,15]. Fis1p has served as a valuable substrate for investigating the principles that govern TA trafficking [16,17,18,19,20]. The Fis1(TA) consists of a hydrophobic stretch of amino acids, predicted to form an alpha-helix, followed by positive charges that promote both insertion and organelle specificity [16,19,21]. Although cytosolic chaperones and other factors may promote the delivery of Fis1p to organelles [22], most evidence suggests that the ultimate insertion of Fis1p into membranes, as well as the insertion of several other tail-anchored proteins at mitochondria, need not be catalyzed by any translocation machinery [17,18,23,24]. However, a recent report has suggested that Fis1p insertion at the mitochondrial surface occurs with the help of the MIM import complex [25], calling into question a model in which Fis1p can spontaneously insert at its destination membranes.

Proteins integrated into membranes fall into three classes: bitopic (transmembrane), monotopic (associated with membrane, but not passing through the lipid bilayer), and polytopic (passing through the membrane several times) [26]. Determination of the topology of Fis1(TA) at lipid bilayers should be informative regarding whether a translocon might be required for organellar insertion.

In this study, we have used atomistic and coarse-grained molecular dynamics simulations to investigate the behavior of the Fis1(TA) at a lipid bilayer mimicking the surface of mitochondria. The interaction of the peptide with the membrane and its response to highly different chemical environments throughout this process required us to use a combination of enhanced simulation techniques to address each step separately. First, we investigated the insertion of the peptide via atomistic simulated annealing runs. Next, to obtain the equilibrium secondary structure inside the membrane, we used atomistic replica exchange simulations. Finally, in order to obtain the free energy for adsorption and transition from monotopic to bitopic orientation within the membrane, we used the coarse-grained MARTINI model in metadynamics simulations coupled with Hamiltonian replica exchange moves.

Our results suggest that the Fis1(TA) can be inserted in a stable, monotopic conformation, supporting the scenario in which this protein is anchored to membranes without the use of a translocation machinery. Our analysis also revealed that Fis1(TA) transition to the bitopic orientation should be inhibited by the presence of a highly charged/polar carboxyl-terminus and linkage, at the TA amino-terminus, to the Fis1 cytosolic domain.

## 2. Materials and Methods

We define the Fis1(TA) peptide as residues 129–155 of the full-length polypeptide. This 27-residue-long region (-LKGVVVAGGVLAGAVAVASFFLRNKRR-COOH) is necessary and sufficient for mitochondrial targeting [10,27]. The position and orientation of the Fis1(TA) peptide with respect to the membrane was quantified by two metrics, *r* and θ. First, the helix axis was defined by using the vector connecting the center-of-mass of the residues 132–134 and 147–149, as these residues remain predominantly in the alpha helix conformation. The orthogonal distance from the center-of-mass of these two groups to the membrane center-line was used to define *r*. The tilt-angle, θ, was defined as the angle between the helical axis and the z-axis in the simulation box.

### 2.1. Atomistic Simulations

All simulations were performed with the GROMACS 2018 simulation package [28]. The CHARMM36 force field for lipids [29] and CHARMM36m all-atom force field for proteins [30] in combination with TIP3P water model [31] were used for the atomistic system. CHARMM-GUI input generator [32,33] was used to assemble the peptide and the lipid membrane system inside a hexagonal simulation box, and also to gather the force field parameters.

The leap-frog algorithm [34] was used with a time step of 2 fs. The Verlet cutoff-scheme with $r_{vdw}=r_{coulomb}=1.2$ nm, as well as force switching to zero from 1.0 to 1.2 nm for VdW forces, were deployed. The particle-mesh Ewald method [35] with a Fourier grid spacing of 0.1 nm was used for electrostatic interactions. H-bonds were constrained by using the LINCS algorithm [36].

Temperature was controlled by velocity-rescaling algorithm [37] with a time constant of 0.1 ps for coupling. Pressure coupling was achieved by Berendsen barostat [38] for equilibration and Parrinello–Rahman [39] for production runs. Semi-isotropic pressure coupling was used to equilibrate the area per lipid. For both in-plane (x-y) and out-of-plane (z) directions reference pressure, compressibility and time constants were set to 1.0 bar, 4.5 × 10${}^{-5}$ bar${}^{-1}$ and 2.5 ps, respectively. Trial runs with a homogeneous DOPC bilayer with the same settings resulted in an average area per lipid of 0.67 nm${}^{2}$, in agreement with earlier simulations and experimental results [40].

Lipid composition of the leaflets, as listed in Table 1, was chosen according to the data in Simbeni et al. [41]. In total, the system contained 200 lipids and 12,000 water molecules. A total of 63 Na${}^{+}$ and 14 Cl${}^{-}$ ions were added to the system to neutralize the system and obtain a salt concentration of 0.15 M.

The Dictionary of Protein Secondary Structure [42] was used for secondary structure assignment. Molecular graphics were produced by VMD [43] and plots were produced with the Gnuplot 5.2 package [44].

#### 2.1.1. Simulated Annealing

Application of simulated annealing (SA) to a multi-component system such as a lipid membrane with a peptide and water as solvent is rather challenging. In order to enforce the peptide to explore different conformations and orientations, with minimal disruption on the membrane and the solvent, we applied the simulated tempering to the peptide only. Within a given cycle during the simulated annealing experiment, the temperature for the peptide was raised from 298 to 800 K in 2.5 ns, kept at 800 K for 2.5 ns, reduced back in 2.5 ns and kept at 298 K for another 12.5 ns. The temperature for the rest of the system was kept constant at 298 K. This allowed us to force the peptide to explore different conformations in and out of the membrane, while minimally disrupting the lipid bilayer and the solvent. For each SA run, the system was simulated for 1000 ns in total. The peptide was initially placed in a random conformation above the membrane in water. Further details of the simulated annealing procedure are provided in Appendix A.

#### 2.1.2. AA-REX

The atomistic replica exchange simulation (AA-REX) [45,46], was initialized with the Fis1(TA) peptide buried close to the upper leaflet, with the center-of-mass of the peptide 1.2 nm below the phosphate groups of the upper leaflet of the membrane. Eighty replicas were used to cover a temperature range of 298 to 471.05 K, where the temperature of each replica was determined by using the web server of Patriksson and van der Spoel [47]. An exchange between neighboring replicas was attempted every 2 ps according to the Metropolis criterion. An average exchange probability of 0.185 was observed. The convergence of the simulations were checked by investigating the secondary structure evolution of the peptide at different temperatures and in different replicas and the results are provided as Appendix A. In addition, in Appendix A, a representative movie demonstrating the conformational dynamics can be seen. Overall, the AA-REX simulation is capable of enforcing conformational sampling for the peptide independent of the initial structure. However, the conformations sampled remain limited to the monotopic orientation; the bitopic orientation was not observed in this simulation.

### 2.2. MARTINI Coarse-Grained Simulations

For coarse-grained simulations, the MARTINI force field [48] was used. The secondary structure of Fis1(TA) was assigned inline with the AA-REX results, with the residues spanning G131 through L150 fixed as helix, and the rest left as random-coil. MARTINI scripts “martinize.py” and “insane.py” were used to prepare the system, and we used the recommended configuration file that is available in MARTINI website [49].

The number and composition of the lipid molecules were chosen identically to the atomistic composition. For every lipid molecule, 25 MARTINI water molecules were added to the simulation box. Na+ and Cl- ions were added to the system both to neutralize the access charge from the peptide and the lipids and to obtain a salt concentration of 0.15 M.

For facilitating the transitions between the monotopic and bitopic orientations, we carried out well-tempered metadynamics simulations with Hamiltonian replica exchange swaps [50,51]. Two collective variables (CV) were used with the metadynamics simulation: tilt-angle (θ) and the orthogonal distance of the peptide from the membrane center (*r*). The metadynamics approach implemented in PLUMED 2 [52] deposits n-dimensional Gaussians, where n is the number of CVs used, with an aim to allow the molecule of interest to overcome high energy barriers. The height of the Gaussian bias was set to 0.50, and the width was set to 0.05 and 0.2 for angle and distance collective variables, respectively. Bias was added every 5000 steps, and biasfactor was chosen as 10.

In both atomistic simulations and trial runs with MARTINI, we observed that the monotopic and bitopic orientations of Fis1(TA) are separated by rather high energy barriers, and even the added bias provided by well tempered metadynamics could not promote transitions. Therefore, we combined well-tempered metadynamics with a Hamiltonian replica exchange-based approach. The wild-type Fis1(TA) was mutated in twelve steps to a highly hydrophobic sequence that would allow transitions from monotopic to bitopic orientation. A total of 78 runs (twelve mutants plus the wild-type and six parallel runs for each) were used to facilitate the sampling of the free-energy surface. For each of the six parallel runs initial condition of the peptide was altered between six configurations: bitopic with amino-terminal in upper/lower leaflet; monotopic in upper/lower leaflet; above and below the membrane in bulk water. Our aim was to enhance transitions between monotopic and bitopic orientations in replicas by using the hydrophobic sequence, wherein both termini lose their charged/polar character gradually, and then “walk” these transitions toward the replica with the wild-type Fis1(TA). The analysis provided is based on the replica that corresponds to the wild-type Fis1(TA).

## 3. Results

We began our analysis by investigating the interaction between Fis1(TA) and a model bilayer mimicking the *S. cerevisiae* mitochondrial outer membrane composition [41]. Initial trials with standard molecular dynamics revealed that even though the peptide spontaneously attaches to the water/membrane interface with its charged termini, the local energy barriers are high enough to inhibit insertion into the membrane (results not shown).

In order to overcome these energy barriers, we utilized the simulated annealing (SA) procedure described in the Methods section. The peptide, initially placed in the aqueous environment, adsorbs onto the membrane by using the five charged/polar residues found at its carboxyl-terminus with (Figure 1a). In the figure, the hydrophobic and charged/polar residues shown in licorice representation are colored white and blue, respectively. The lipids in the upper leaflet are shown in transparent gray ball-and-stick representation and the water molecules are not shown for clarity. As the peptide slowly inserts itself into the membrane, the amino-terminus of the Fis1(TA) also adsorbs. Being attached from both termini to the membrane, the peptide slowly penetrates the lipid environment, as revealed by the orthogonal distance from the center-of-mass of the peptide to the membrane center-line (*r*) (Figure 1c). The water/membrane interface (defined as the point where the lipid density profile equals 500 kg/m${}^{3}$) is shown in the figure with a dashed blue-line. The density profiles for lipids and water (Figure 1c right side) show that at this depth the charged residues of the peptide can still interact with both the solvent and the charged head groups of the lipids. After 520 ns, the peptide folds into an extended α-helix conformation inside the membrane in a monotopic orientation (Figure 1b), as revealed by the secondary structure analysis of the molecule (Figure 1d).

For testing the validity of the SA procedure, we have performed eleven SA runs by using different initial random seeds for both the wild-type Fis1(TA) and two mutants A144D and L139P. Previously, these mutants were shown to drastically reduce the binding activity of the Fis1(TA) to membranes [19]. Figure 2 shows the orthogonal distance from the peptide center-of-mass to the membrane center line (a) and the number of residues in α-helix conformation (b) during the SA runs for both the wild-type and the mutants. Individual runs for each of the three systems show a diverse behavior; however, a general pattern that distinguishes the wild-type from the mutants can be clearly observed. All three systems adsorb to the membrane via their charged/polar carboxyl-termini. However, unlike the wild-type Fis1(TA), the mutants A144D and V139P cannot penetrate with sufficient depth into the membrane, and therefore cannot acquire a stable secondary structure as indicated by their lower α-helix propensity (see Appendix A for detailed analysis of the SA runs for each case.) A closer inspection of A144D and L139P mutants demonstrate that these two mutants display different mode of actions. The A144D mutant, due to an extra charge introduced in its hydrophobic segment, negatively affects the Fis1(TA)’s insertion into the membrane. However, if it is inserted, which is a rarer event compared to the wild-type Fis1(TA), the peptide can still adopt an α-helix. On the other hand, the L139P mutant mainly disrupts the helix propensity of the hydrophobic block, consistent with the helix-breaking character of proline residues. These results indicate that the lower membrane-binding ability of the A144D and L139P Fis1(TA) mutants seen in vivo [19] is associated with their inability to penetrate deep inside the membrane and to acquire a stable conformation.

Even though the monotopic orientation was stable for wild-type Fis1(TA) in the 1000 ns SA runs, one cannot rule out the possibility that, following insertion, the Fis1(TA) might reach an unassisted bitopic orientation of lower energy. Inside the membrane, the conformational dynamics of the peptide slow significantly. As a result, our SA approach is not sufficient for obtaining the equilibrium secondary structure of the peptide and its preferred orientation inside the membrane.

Consequently, we performed a μs-long atomistic replica exchange molecular dynamics (AA-REX) simulation [53,54] on the Fis1(TA) peptide. The Fis1(TA) was initially placed inside the lipid bilayer as a fully extended alpha-helix in the monotopic orientation, 1.2 nm below the lipid/water interface. Please see the Methods section for further details of the AA-REX simulation.

At the AA-REX target temperature of 298 K, the peptide retained a predominantly alpha-helical conformation and a monotopic orientation, as depicted in the representative snapshot in Figure 3a. Residue-based secondary structure propensity analysis of the 298 K replica’s trajectory revealed that, except for the five charged and/or polar residues at the carboxyl-terminus, the peptide displays a strong tendency towards alpha-helicity (Figure 3b). The helical, hydrophobic portion of the peptide is submerged approximately 0.7 nm below the lipid/water interface, whereas the non-helical, charged carboxyl-terminus extends towards this interface (Figure 3c). The tilt angle, defined here as the angle between the helix-axis and the membrane normal, clearly indicates (Figure 3d) that the peptide maintains a monotopic orientation. Even though the AA-REX simulation is effective in revealing the equilibrium secondary structural preference of the peptide, we did not observe any transitions from the monotopic to bitopic orientation during the μs-long simulation.

The size of the AA-REX system hampers a thorough exploration of the possible transitions from monotopic to bitopic orientation and any associated free energy changes. Therefore, to further investigate the insertion free energy and preferred orientation of the Fis1(TA) in the membrane, we performed well-tempered metadynamics simulations coupled with Hamiltonian replica exchange [50,51] by using the coarse-grained MARTINI force field [48,55]. The MARTINI force field has been widely used for studying the conformational dynamics of proteins with the lipid bilayer [56,57,58]. As the MARTINI force field does not allow conformational changes, the secondary structure of the peptide was assigned according to the AA-REX results provided in Figure 3b. The helical tilt angle (θ) and the orthogonal distance from the center-of-mass of the peptide to the membrane center line (*r*) were used as collective variables for the metadynamics simulation.

As indicated by the earlier results, the charged termini prevent the transitions between monotopic and bitopic orientations for the Fis1(TA). In order to facilitate this transition, we have created twelve separate mutants, which gradually turn into a completely hydrophobic sequence. Next, we ran metadynamics simulations for all thirteen systems (wild-type and 12 mutants) in parallel, and used the Hamiltonian replica exchange method with swaps between neighboring replicas every 5000 steps during the simulation. Furthermore, for each system six different simulations were set up with the peptide initially placed in different orientations: monotopic in upper/lower leaflet, bitopic with carboxyl-termini in upper/lower leaflet, and in water above/below the membrane. The combined usage of metadynamics and Hamiltonian replica exchange in this set of 78 simulations [13 systems (wildtype and mutants) × 6 initial conditions] allowed us to sample the conformational space in all systems (please see Appendix A for the analysis of convergence).

The resulting free energy surface for the wild-type Fis1(TA) is shown in Figure 4a. Four distinct minima are observed in the free energy surface: the two monotopic orientations (labeled 1 and 3) and the two bitopic orientations (labeled 2 and 4) on either side of the membrane. Representative conformations of the peptide within the membrane for each minima are shown in Figure 4b. The pathway from 1-to-2 (or 3-to-4, considering the symmetric membrane model) represents a monotopic-to-bitopic transition via the amino-terminus, whereas the pathway from 1-to-4 (or 3-to-2) represents a transition via the carboxyl-terminus. As revealed by the free-energy surface, transitions to the bitopic orientation are prohibited by high energy barriers (40 kJ/mol for the amino- and 60 kJ/mol for the carboxyl-terminus).

The energy difference between the monotopic and bitopic orientations is within the margin of error of the free-energy analysis. Due to the parallel alignment of the helix with respect to the membrane normal, the bitopic orientation disrupts the lipid molecules much less compared to the monotopic orientation. This allows for higher conformational freedom for the peptide and leads to a larger basin in the free-energy surface.

The metadynamics simulations yield the insertion free energy from water to the membrane as 80 kJ/mol, in agreement with the insertion for wild-type Fis1(TA) observed in SA simulations. When compared to the AA-REX simulation, the peptide’s center-of-mass in the monotopic orientation is closer to the lipid/water interface (≈0.25 nm below the interface) in the MARTINI simulation.

As revealed by the metadynamics simulations, an unassisted transition from the monotopic orientation to bitopic orientation is highly unlikely. The transition via the carboxyl-terminus is blocked due to the four positive charges. The transition via amino-terminus is slightly more favorable, however this pathway would also be impossible in the context of a full-length Fis1 polypeptide.

## 4. Discussion

Previous results suggest that the Fis1(TA) does not require any specific translocation machinery for its membrane insertion [17,18]. Indeed, Fis1p is integrated at mitochondrial outer membranes treated with protease to destroy receptors and translocation machinery, and Fis1p can even be inserted into bare liposomes. Moreover, Fis1p plays a role in the division of both peroxisomes and mitochondria [9,10,11,12], although no translocation machinery is known to be shared between these two organelles, further suggesting a translocon-independent mode of insertion.

Our atomistic and coarse-grained simulations, performed by using a model mitochondrial outer membrane, suggest that the Fis1(TA) can be stably inserted monotopically into only one leaflet of a lipid bilayer in the absence of a translocation machinery. Fis1(TA)’s charged carboxyl-terminus, which is likely to be responsible for its initial adsorption to mitochondrial membrane, seems to restrict an unassisted monotopic-to-bitopic transition. In this regard, our findings are consistent with the literature, as other TA peptides with an expanded charged termini have been known to favor a monotopic orientation [59,60,61]. We speculate that the dual localization of Fis1p at mitochondria and peroxisomes is likely to be driven by the lipid environments found at those membranes, and specific lipids that might promote Fis1p association with its target organelles are a subject of current investigation.

Because we have only studied the interaction of a single Fis1(TA) peptide with the membrane, we cannot rule out the transition to a bitopic orientation via correlated interaction of multiple peptides with the membrane, which was proposed as a viable mechanism for highly charged cationic cell penetrating peptides [62]. Interaction of multiple Fis1(TA)s with a lipid membrane will be the subject of a future study.

## Figures and Tables

**Figure 1 membranes-12-00752-f001:**
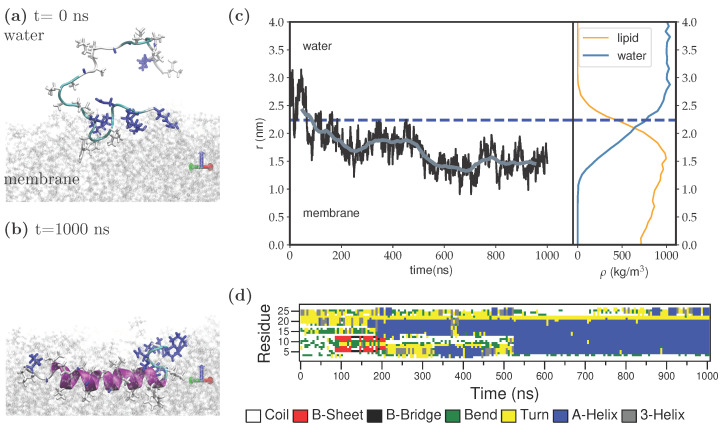
Wild-type Fis1(TA) adsorbs onto the lipid membrane via its carboxyl-terminus, then folds into an alpha-helical structure found within the membrane in the monotopic orientation. Snapshots for (**a**) the first contact with the membrane and (**b**) the final helical structure in monotopic orientation. (**c**) Orthogonal distance from the center-of-mass of the peptide to the membrane center-line (left) and the density profile for lipids and water above the membrane center-line (right). The water/membrane interface is shown with the dashed blue line. (**d**) Secondary structure evolution during insertion into the membrane.

**Figure 2 membranes-12-00752-f002:**
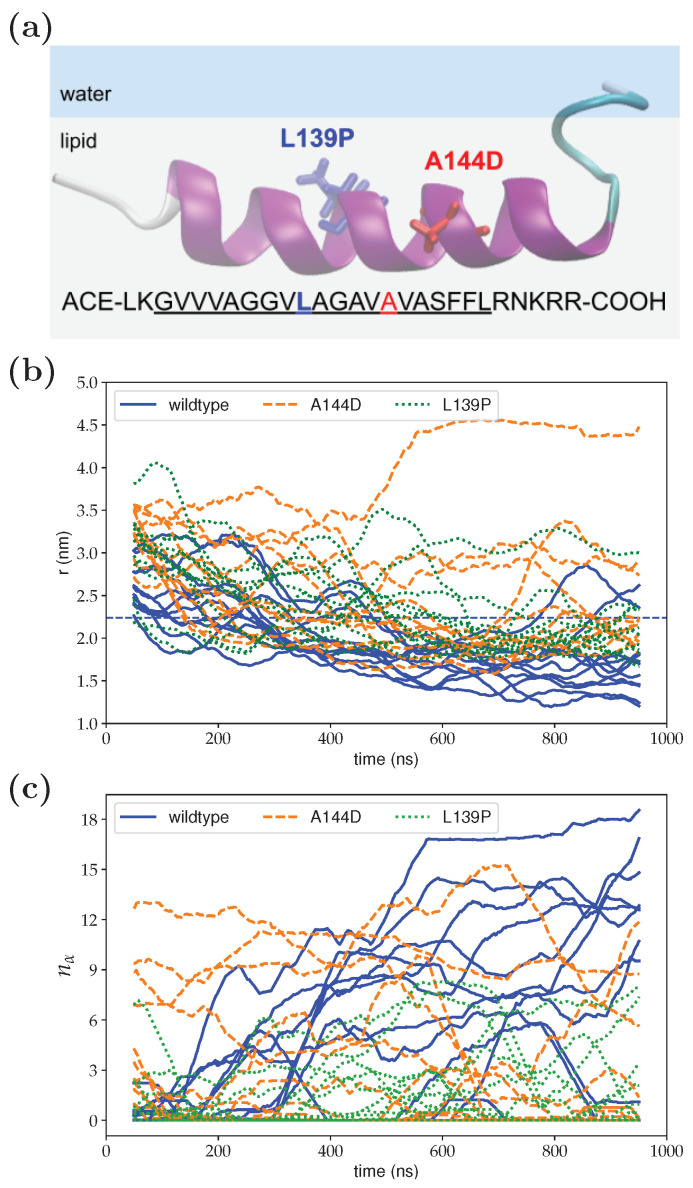
Comparison of the wild-type Fis1(TA), A144D and L139P mutations via eleven separate SA runs for each. (**a**) Graphical representation of the molecules; (**b**) the orthogonal distance from the peptide center-of-mass to the membrane center line; (**c**) the number of residues in α-helix conformation. Wild-type Fis1(TA), A144D, and L139P results are shown with solid blue, dashed orange and dotted green lines, respectively.

**Figure 3 membranes-12-00752-f003:**
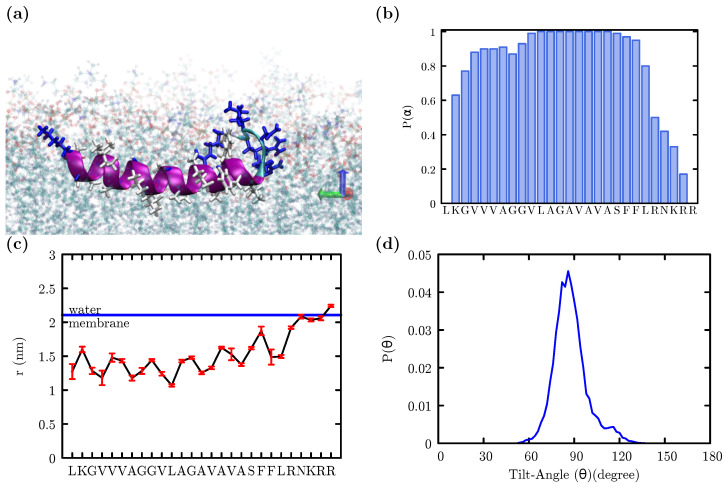
Analysis of AA-REX results. (**a**) Fis1 TA adopts a dominantly α-helical structure within the membrane. (**b**) Sequence specific α-helix propensity. (**c**) Average depth of the residues with respect to membrane/water interface. (**d**) Angle between the helix axis and the membrane normal.

**Figure 4 membranes-12-00752-f004:**
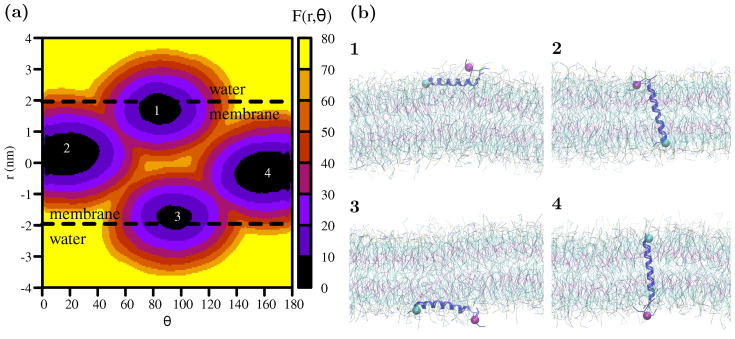
(**a**) The free-energy surface for the Fis1(TA) peptide’s insertion into the membrane. The color scheme represents the relative free energy F(r,θ) with respect to the minimum as indicated by the color bar. Four distinct minima represent the monotopic (1 and 3) and bitopic (2 and 4) orientations of the peptide in the membrane. (**b**) Representative snapshots of the peptide in each case. Amino-terminus and carboxyl-terminus are marked by cyan- and purple-colored spheres.

**Table 1 membranes-12-00752-t001:** Composition and charge of the lipids in the membrane, obtained from [41].

Lipid	Percentage	Charge
DOPC (PC)	46%	0
DOPE (PE)	33%	0
POPI (PI)	10%	−1
Cardiolipin	6%	−2
DOPA (PA)	4%	−1
DOPS (PS)	1%	−1

## Data Availability

The authors declare that the data supporting the findings of this study are available within the article and its Appendix A. The raw MD simulation trajectories can be obtained from the corresponding author (Mehmet Sayar) upon request.

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
