# Peer review of "Modeling Adsorption, Conformation, and Orientation of the Fis1 Tail Anchor at the Mitochondrial Outer Membrane"

_membranes, 2022, doi:10.3390/membranes12080752_

Round 1

Reviewer 1 Report

The manuscript entitled 'Modeling Adsorption, Conformation, and Orientation of the Fis1 Tail Anchor at the Mitochondrial Outer Membrane' is a good piece of paper. The paper has properly discussed relevant literature and related studies. The paper has taken a sound scientific approach and shows appropriate graphs for visualization. Finally, a convincing conclusion is obtained.

However, I would like to have such following suggestions to improve the paper:

1.I’m concerned about the incomplete literature.

2.The introduction's third paragraph, "Integral membranes …", is a weird transition. Please rewrite it to make it more natural.

3.Pictures can be drawn more nicely. Figure1(c). The lines divide the numbers 0.0 and 4.0.

4.Figure 4(a). Please indicate the physical meaning of the right y coordinate.

Author Response

We would like to thank both reviewers for a detailed reading of our manuscript and their suggestions and comments. Our replies to their comments and the list of modifications we have done on the manuscript are listed below. 

1.I’m concerned about the incomplete literature.

We have updated the literature in the manuscript, and we believe that it's current state is comprehensive and adequate.  If the reviewer can guide us towards any specific paper  that is relevant to our study, we would be happy to consider it. 

2.The introduction's third paragraph, "Integral membranes …", is a weird transition. Please rewrite it to make it more natural.

We have rewritten this paragraph as follows:

"Proteins integrated into membranes fall into three classes: Bitopic (trans-membrane), monotopic (associated with membrane, but not passing through the lipid bilayer), and polytopic (passing through the membrane several times) \cite{Blobel1980}. Determination of the topology of Fis1(TA) at lipid bilayers should be informative regarding whether a translocon might be required for organellar insertion."

3.Pictures can be drawn more nicely. Figure1(c). The lines divide the numbers 0.0 and 4.0.

We have corrected the figure by adding a y2axis (and removing the yaxis) for the right hand side plot to avoid the overlap. 

4.Figure 4(a). Please indicate the physical meaning of the right y coordinate.

The colorbar represents the free-energy profile for the Fis1TA peptide. We have added a label (F(r,theta)) to the figure and also updated the figure caption. 

In addition to the above we have also corrected figure 2 caption. 

Reviewer 2 Report

The study is about modeling adsorption, conformation, and orientation of the Fis1 tail anchor at the mitochondrial outer membrane. The authors performed atomistic and coarse-grained molecular dynamics stimulations to analyzed the adsorption, conformation, and orientation of the Fis1 tail anchor at the mitochondrial outer membrane. Their findings support stable insertion at the mitochondrial outer membrane in a monotopic rather than a bitopic configuration.

minor concern:

Do the authors have an explanation to why an unassisted transition from the monotopic orientation to bitopic are less likely to occur?

Author Response

We would like to thank both reviewers for a detailed reading of our manuscript and their suggestions and comments. Our replies to their comments (marked in blue color) and the list of modifications we have done on the manuscript are listed below. 

Do the authors have an explanation to why an unassisted transition from the monotopic orientation to bitopic are less likely to occur?

The free energy profile (Fig 4) as obtained by the Martini simulations indicate the presence of a high energy barrier, which prevents monotopic-to-bitopic transition  (40 kJ/mol for the amino- and 60 kJ/mol for the carboxyl-terminus). This is discussed in the manuscript's Results section (lines 257-261 and 273-277).